# Genetic analysis of a potato (*Solanum tuberosum* L.) breeding collection for southern Colombia using Single Nucleotide Polymorphism (SNP) markers

**Jhon A. Berdugo-Cely** [1] *, **Carolina Martínez-Moncayo** [2], **Tulio César Lagos-Burbano** [3]

1 Corporación Colombiana de Investigación Agropecuaria-AGROSAVIA, Centro de Investigación Turipaná, Montería, Cereté, Colombia, 2 Grupo de Investigación en Producción de Frutales Andinos, Universidad de Nariño, Pasto, Colombia, 3 Facultad de Ciencias Agrícolas, Universidad de Nariño, Pasto, Nariño, Colombia

* jberdugo@agrosavia.co

**Data Availability Statement:** All relevant data are within the manuscript and its Supporting Information files.

## Abstract

Detailed knowledge on genetic parameters such as diversity, structure, and linkage disequilibrium (LD) and identification of duplicates in a germplasm bank and/or breeding collection are essential to conservation and breeding strategies in any crop. Therefore, the potato genetic breeding collection at the Universidad de Nariño in Colombia, which is made up of diploid and tetraploid genotypes in two of the more diverse genebanks in the world, was analyzed with 8303 single nucleotide polymorphisms (SNP) from SolCAP version 1. In total, 144 genotypes from this collection were analyzed identifying an 57.2% of the polymorphic markers that allowed establishing two and three subpopulations that differentiated the diploid genotypes from the tetraploids. These subpopulations had high levels of heterozygosity and linkage disequilibrium. The diversity levels were higher in the tetraploid genotypes, while the LD levels were higher in the diploid genotypes. For the tetraploids, the genotypes from Peru had greater diversity and lower linkage disequilibrium than those from Colombia, which had slightly lower diversity and higher degrees of LD. The genetic analysis identified, adjusted and/or selected diploid and tetraploid genotypes under the following characteristics: 1) errors in classification associated with the level of ploidy; 2) presence of duplicates; and 3) genotypes with broad genetic distances and potential use in controlled hybridization processes. These analyses suggested that the potato genetic breeding collection at the Universidad de Nariño has a genetic base with a potential use in breeding programs for this crop in the Department of Nariño, in southern Colombia.

## Introduction

The potato (*Solanum tuberosum* L.) is the most important non-cereal crop in the world, with more than 368 million tons produced from approximately 4,000 varieties grown on 17.5 million hectares [1, 2]. This crop is key to food security because of its high nutritional value provided by carbohydrates, proteins, fibers, minerals and vitamins [3, 4]. The increasing world

**Funding:** The resources for the development of this research were provided by the Ministerio de Ciencia y Tecnología de Colombia (Minciencias) and the Gobernación de Nariño through the Fondo de Ciencia, Tecnología e Innovación del Sistema General de Regalías, with the approval of the project "Improvement technological and productive of the potato system in the Department of Nariño", identified with code BPIN No. 2014000100022. The execution of this project was carried out between the Universidad de Nariño and AGROSAVIA through the macro agreement 480-15. The hours spent by researcher Jhon A. Berdugo-Cely MSc. for the development of this study were provided by AGROSAVIA through Variable Transfer (TV) 2019. The funders had no role in study design, data collection and analysis, decision to publish, or preparation of the manuscript.

**Competing interests:** The authors have declared that no competing interests exist.

population means the demand for food will increase, requiring the constant development of improved cultivars to meet the needs of consumers, producers and processors, who require potato genetic materials with: 1) better taste and high nutritional value; 2) higher production; 3) resistance to pests and diseases; and 3) low content of reducing sugars and starch, among other compounds [5]. The genetic variability of potato genotypes with potential use in genetic breeding processes for the development of new cultivars must be identified and explored.

In 2018, Colombia ranked 23rd for global potato producers with three million tons cultivated on 133 thousand hectares [2], of which 399 thousand tons of potatoes (tetraploids) and 16 thousand tons of "Criollas" potatoes (diploids) were produced in the Department of Nariño, which is the third largest potato producer nationwide [6]. Although the Pastusa Suprema, Diacol Capiro, Parda Pastusa, Superior and Criolla varieties are the better known and more cultivated ones in the Department of Nariño [7], new genotypes adapted to the agroecological conditions of this region with desirable characteristics for consumers and the industrial use are needed. The project "Technological and productive improvement of the potato system in the Department of Nariño" [8] aims to identify outstanding genetic materials for conditions of abiotic stresses such as a water deficit and low fertilization levels between 2400 and 3000 meters above sea level for southern Colombia. For this, a genetic breeding collection was created consisting of 506 potato genotypes, which include materials with multiple collection origins, mainly the germplasm bank at the International Potato Center (CIP) of Peru (54), the Central Colombian Collection (CCC) (76), and the Universidad de Nariño of Colombia (376). This breeding collection is undergoing a morpho-agronomic evaluation for the selection of promising genotypes for possible use as parents in the potato genetic breeding program at the Universidad de Nariño or as candidate genetic materials for possible registration as new varieties from the Andean region of southern Colombia. This selection could be carried out with genomic tools for the genetic characterization of this collection using molecular markers.

Potato genetic diversity is mainly estimated with morphological and physiological characteristics, such as plant architecture, resistance to diseases, and shape and color of flowers and tubers. However, many of these characteristics are affected by the environment [5, 9]. Therefore, methodologies such as molecular markers, which are not affected by the environment, are currently used for the estimation of genetic variability in plant species because of their neutrality, Mendelian inheritance and ease of detection in any tissue and growth stage in plants [5]. In potatoes, different types of DNA markers have been widely implemented for genetic analysis, including random amplified polymorphic DNA (RAPD), microsatellites (SSR) [10], polymorphic length restriction fragments (RFLP) [11], and amplified fragment length polymorphism (AFLP) [12] but single nucleotide polymorphisms (SNP) are used most often [13].

Molecular markers such as SNPs are point variations in nucleotides throughout genomes and have been used for: 1) analysis of genetic diversity; 2) phylogenetic analysis; 3) identification of genes with agronomic importance with Quantitative Traits Loci (QTLs) and Genome-Wide Association Studies (GWAS) mapping; 4) Marker-Assisted Selection (MAS); and 5) varietal identification, among other applications [14]. High-throughput SNP genotyping platforms based on hybridization and fluorescence have been developed for potatoes, where 8K [15] and 20K [13] SNParrays are available. The 8K array contains a subset of 8303 SNPs selected from a set of more than 69 thousand markers identified between transcriptomics and EST (Expressed Sequence Tag) data for six North American cultivars (Bintje, Kennebec, Premier Russet, Shepody, Snowden, and Atlantic) [16]. This matrix has been used to study the genetic diversity of potato germplasm collections of European origin and from North and South America [17–19]. Additionally, SNPs have been used to infer the phylogenetic relationship of some species of *Solanum* sect. Petota [20] and to identify gene candidates of economic importance with QTLs [21, 22], and GWAS [23, 24] mapping.

The objective of the present study was to analyze 144 *Solanum tuberosum* genotypes in the potato genetic breeding collection at the Universidad de Nariño in southern Colombia at the genetic level with single nucleotide polymorphisms to establish the: 1) diversity; 2) genetic structure; and 3) linkage disequilibrium; and 4) identify candidate genotypes for duplicates and/or potential use in controlled hybridization processes.

## Materials and methods

### Plant material

This study included 144 potato genotypes from the genetic breeding collection at the Universidad de Nariño in southern Colombia (Table 1). The genetic materials in this collection belong from multiple germplasm bank origins: the International Potato Center (CIP) of Peru, the Colombian Central Collection (CCC), and the Universidad de Nariño of Colombia and were selected based on the following criteria: 1) yield, 2) industrial potential, and 3) tolerance to *Phytophthora infestans*. The working collection is preserved under *in-vitro* and field conditions. For the former, the collection is kept in the plant tissue culture laboratory of the Grupo de Investigación en Producción de Frutales Andinos located at the Universidad de Nariño at 01˚12'13"LN, 77˚15'23"LW and 2540 masl, with a photoperiod of 12/12 hours light/dark; 20 explants of each genetic material are preserved in Murashige & Skoog (1962) culture medium. For field conditions, this collection was sown on the Granja Experimental La Botana of the Universidad de Nariño in plots with 10 clones per introduction. This farm is located on the high plateau of Pasto at 01˚09'12"LN, 77˚18'31"LW and 2820 masl, with an average temperature of 13˚C and 970 hours of sun/year, rainfall at 803 mm/year and 82% relative humidity.

### DNA extraction

For each of the 144 potato materials from the genetic breeding collection at the Universidad de Nariño, young leaves were collected for each genotype grown under *in-vitro* conditions, from which DNA was isolated using an Extract-N- Amp™ Plant PCR Kit from Sigma-Aldrich, Germany. The quality of the DNA was verified with visualization in 1% agarose gels stained with ethidium bromide (0.5ng/mL), while the DNA concentration was estimated with spectrophotometry using NanoDrop 2000 (Thermo Fisher Scientific, Wilmington, USA). Finally, the DNA was diluted to a final concentration of 100 ng/μL and stored at -20˚C until genotyping.

### Genotyping and SNP selection

The genotyping of the potato genetic breeding collection at the Universidad de Nariño was carried out with an 8K matrix [15] from Infinium technology; the beadcheaps were read with an Illumina HiScan SQ (Illumina, San Diego, CA) at the Corporación Colombiana de Investigación Agropecuaria—AGROSAVIA in Tibaitatá research center at Mosquera—Colombia. The fluorescence intensities were extracted from the GenomeStudio program (Illumina, San Diego CA) to assign genotypes to each locus (0, 1, 2, 3, 4), which was carried out with the Fit-Tetra library [25] in the R program [26]. The markers that could not be determined or that were monomorphic were discarded; the remaining markers were subjected to a new filter, with more than 20% and 5% of data lost at the population level and for the Minimum Allele Frequencies (MAF), respectively (S1 Table).

### Structure and genetic diversity

The analysis of the population structure of the potato genetic breeding collection at the Universidad de Nariño used a tetraploid model (0, 1, 2, 3, 4) with two strategies: A) A Bayesian

**Table 1. List of genotypes from the potato breeding collection of the Universidad de Nariño analyzed in this study.**

| Code | Genotype | Code or vulgar name | Ploidy by PD[1] | Country | Department | Pop_K2 [2] | Pop_K3 [3] | Ploidy by GS[4] |
|------|----------|---------------------|-----------------|---------|------------|------------|------------|-----------------|
| Ext1 | UdenarStCr16 | C.I.O 31.12 (Ratona morada) | D | C | Nariño | S_Nariño_1 | P_Nariño_1 | D |
| Ext12 | UdenarStCr35 | Ñoña | D | C | Nariño | S_Nariño_1 | P_Nariño_1 | D |
| Ext13 | UdenarStCr47 | Mambera M1 | D | C | Nariño | S_Nariño_1 | P_Nariño_1 | D |
| Ext148 | UdenarStCr178 | CIP 703548 | D | P | U | S_Nariño_1 | P_Nariño_1 | D |
| Ext154 | UdenarStCr179 | CIP 703572 | D | P | U | S_Nariño_1 | P_Nariño_1 | D |
| Ext158 | UdenarStCr182 | CIP 703567 | D | P | U | S_Nariño_1 | P_Nariño_1 | D |
| Ext16 | UdenarStCr55 | Ratona La Cocha Reserva Cristales | D | C | Nariño | S_Nariño_1 | P_Nariño_1 | D |
| Ext162 | UdenarStCr176 | CIP 703594 | D | P | U | S_Nariño_1 | P_Nariño_1 | D |
| Ext17 | UdenarStCr23 | Chaucha pura | D | C | Nariño | S_Nariño_1 | P_Nariño_1 | D |
| Ext197 | UdenarStCr180 | CIP 703546 | D | P | U | S_Nariño_1 | P_Nariño_1 | D |
| Ext2 | UdenarStCr44 | Dorada | D | C | Nariño | S_Nariño_1 | P_Nariño_1 | D |
| Ext21 | UdenarStGua18 | Unica (Cordoba) | T | C | Nariño | S_Nariño_1 | P_Nariño_1 | D |
| Ext214 | UdenarStCr21 | C.I.O 40.21 (Pacha negra) | D | C | Nariño | S_Nariño_1 | P_Nariño_1 | D |
| Ext216 | UdenarStCr08 | C.I.O 23.4 (Malvaseña) | D | C | Nariño | S_Nariño_1 | P_Nariño_1 | D |
| Ext229 | UdenarStCr17 | C.I.O 34.15 (Tornilla Roja) | D | C | Nariño | S_Nariño_1 | P_Nariño_1 | D |
| Ext241 | UdenarStCr54 | Ratona blanca criolla la cocha M1 flor morada | D | C | Nariño | S_Nariño_1 | P_Nariño_1 | D |
| Ext242 | UdenarStCr140 | Jardinera- 15061382 | D | C | Norte De Santander | S_Nariño_1 | P_Nariño_1 | D |
| Ext243 | UdenarStCr09 | C.I.O 24.5 (Calabera) (Chaucha Negra) | D | C | Nariño | S_Nariño_1 | P_Nariño_1 | D |
| Ext244 | UdenarStCr42 | America | D | C | Nariño | S_Nariño_1 | P_Nariño_1 | D |
| Ext245 | UdenarStCr01 | Andina | D | U | U | S_Nariño_1 | P_Nariño_1 | D |
| Ext246 | UdenarStCr40 | Yema de huevo | D | C | Nariño | S_Nariño_1 | P_Nariño_1 | D |
| Ext248 | UdenarStCr10 | C.I.O 25.6 (Cachuda) | D | C | Nariño | S_Nariño_1 | P_Nariño_1 | D |
| Ext250 | UdenarStCr129 | Argentina Parda- 15061273 | D | C | Boyacá | S_Nariño_1 | P_Nariño_1 | D |
| Ext3 | UdenarStCr57 | Ratona morada M4 | D | C | Nariño | S_Nariño_1 | P_Nariño_1 | D |
| Ext33 | UdenarStCr63 | Uva negra phureja M4 (nn uva negra) | D | C | Nariño | S_Nariño_1 | P_Nariño_1 | D |
| Ext4 | UdenarStCr45 | Kurikinga M1 | D | C | Nariño | S_Nariño_1 | P_Nariño_1 | D |
| Ext40 | UdenarStCr67 | Ratona Roja M5 | D | C | Nariño | S_Nariño_1 | P_Nariño_1 | D |
| Ext42 | UdenarStCr33 | Jardinera | D | C | Nariño | S_Nariño_1 | P_Nariño_1 | D |
| Ext44 | UdenarStCr73 | Ratona Gourmet | D | C | Nariño | S_Nariño_1 | P_Nariño_1 | D |
| Ext45 | UdenarStCr52 | Nacional 2 | D | C | Nariño | S_Nariño_1 | P_Nariño_1 | D |
| Ext46 | UdenarStCr51 | Nacional 1 | D | C | Nariño | S_Nariño_1 | P_Nariño_1 | D |
| Ext47 | UdenarStCr75 | Criolla Nativa | D | C | Nariño | S_Nariño_1 | P_Nariño_1 | D |
| Ext48 | Unknown | Norteña M3 | U | U | U | S_Nariño_1 | P_Nariño_1 | D |
| Ext52 | UdenarStCr50 | Morada Sigifredo M1 | D | C | Nariño | S_Nariño_1 | P_Nariño_1 | D |
| Ext54 | UdenarStCr66 | Nevada pequeña | D | C | Nariño | S_Nariño_1 | P_Nariño_1 | D |
| Ext55 | UdenarStCr62 | Tornilla la cocha Cristales Roberto Jojoa | D | C | Nariño | S_Nariño_1 | P_Nariño_1 | D |
| Ext57 | UdenarStCr68 | Mambera pintada | D | C | Nariño | S_Nariño_1 | P_Nariño_1 | D |
| Ext6 | UdenarStCr39 | Tornilla negra | D | C | Nariño | S_Nariño_1 | P_Nariño_1 | D |
| Ext60 | UdenarStCr74 | C.I.O 32.13 (Ratona negra) | D | C | Nariño | S_Nariño_1 | P_Nariño_1 | D |
| Ext61 | UdenarStCr20 | C.I.O 39.20 (Curipanga) | D | C | Nariño | S_Nariño_1 | P_Nariño_1 | D |
| Ext62 | UdenarStCr69 | Morada NN M5 (nn morada) | D | C | Nariño | S_Nariño_1 | P_Nariño_1 | D |
| Ext67 | Unknown | MAMA RATONA MORADA M3 | U | U | U | S_Nariño_1 | P_Nariño_1 | D |
| Ext68 | UdenarStCr43 | Botella roja | D | C | Nariño | S_Nariño_1 | P_Nariño_1 | D |

*(Continued)*

**Table 1.** (*Continued*)

| Code | Genotype | Code or vulgar name | Ploidy by PD[1] | Country | Department | Pop_K2 [2] | Pop_K3 [3] | Ploidy by GS[4] |
|---|---|---|---|---|---|---|---|---|
| **Ext71** | UdenarStCr60 | Tornilla amarilla | D | C | Nariño | S_Nariño_1 | P_Nariño_1 | D |
| **Ext73** | UdenarStCr61 | Tornilla Blanca M1 | D | C | Nariño | S_Nariño_1 | P_Nariño_1 | D |
| **Ext74** | UdenarStCr64 | Silvania 1 | D | C | Nariño | S_Nariño_1 | P_Nariño_1 | D |
| **Ext8** | UdenarStGua19 | Guata Roja Antigua La Cocha | T | C | Nariño | S_Nariño_1 | P_Nariño_1 | D |
| **Ext104** | UdenarStGua51 | 15062421—SABANERA | T | C | Cundinamarca | S_Nariño_2 | P_Nariño_2 | T |
| **Ext105** | UdenarStCr135 | Visinia- 15061323 | D | C | Boyacá | S_Nariño_2 | P_Nariño_2 | T |
| **Ext11** | UdenarStGua29 | Parda Pastusa Surco 22 M2 | T | U | U | S_Nariño_2 | P_Nariño_2 | T |
| **Ext144** | UdenarStGua58 | CIP 387164.4 | T | P | U | S_Nariño_2 | P_Nariño_2 | T |
| **Ext146** | UdenarStGua96 | CIP 398192.592 | T | P | U | S_Nariño_2 | P_Nariño_2 | T |
| **Ext147** | UdenarStGua55 | CIP 377744.1 | T | P | U | S_Nariño_2 | P_Nariño_2 | T |
| **Ext149** | UdenarStGua87 | CIP 396012.266 | T | P | U | S_Nariño_2 | P_Nariño_2 | T |
| **Ext150** | UdenarStGua79 | CIP 393382.44 | T | P | U | S_Nariño_2 | P_Nariño_2 | T |
| **Ext151** | UdenarStGua90 | CIP 396038.101 | T | P | U | S_Nariño_2 | P_Nariño_2 | T |
| **Ext152** | UdenarStGua84 | CIP 395193.6 | T | P | U | S_Nariño_2 | P_Nariño_2 | T |
| **Ext153** | UdenarStGua57 | CIP 384866.5 | T | P | U | S_Nariño_2 | P_Nariño_2 | T |
| **Ext156** | UdenarStGua98 | CIP 398208.620 | T | P | U | S_Nariño_2 | P_Nariño_2 | T |
| **Ext159** | UdenarStGua76 | CIP 393371.159 | T | P | U | S_Nariño_2 | P_Nariño_2 | T |
| **Ext160** | UdenarStGua69 | CIP 392657.8 | T | P | U | S_Nariño_2 | P_Nariño_2 | T |
| **Ext161** | UdenarStGua53 | CIP 300046.22 | T | P | U | S_Nariño_2 | P_Nariño_2 | T |
| **Ext163** | UdenarStGua78 | CIP 393371.58 | T | P | U | S_Nariño_2 | P_Nariño_2 | T |
| **Ext164** | UdenarStGua89 | CIP 396034.268 | T | P | U | S_Nariño_2 | P_Nariño_2 | T |
| **Ext165** | UdenarStGua94 | CIP 398190.404 | T | P | U | S_Nariño_2 | P_Nariño_2 | T |
| **Ext166** | UdenarStGua100 | CIP 399075.7 | T | P | Peru | S_Nariño_2 | P_Nariño_2 | T |
| **Ext167** | UdenarStGua80 | CIP 394611.112 | T | P | U | S_Nariño_2 | P_Nariño_2 | T |
| **Ext168** | UdenarStGua54 | CIP 300056.33 | T | P | U | S_Nariño_2 | P_Nariño_2 | T |
| **Ext169** | UdenarStGua65 | CIP 391691.96 | T | P | U | S_Nariño_2 | P_Nariño_2 | T |
| **Ext170** | UdenarStGua63 | CIP 391058.175 | T | P | U | S_Nariño_2 | P_Nariño_2 | T |
| **Ext171** | UdenarStGua68 | CIP 392657.171 | T | P | U | S_Nariño_2 | P_Nariño_2 | T |
| **Ext172** | UdenarStGua86 | CIP 395446.1 | T | P | U | S_Nariño_2 | P_Nariño_2 | T |
| **Ext173** | UdenarStGua67 | CIP 392633.64 | T | P | U | S_Nariño_2 | P_Nariño_2 | T |
| **Ext174** | UdenarStGua83 | CIP 395112.32 | T | P | U | S_Nariño_2 | P_Nariño_2 | T |
| **Ext175** | UdenarStGua61 | CIP 391011.17 | T | P | U | S_Nariño_2 | P_Nariño_2 | T |
| **Ext177** | UdenarStGua70 | CIP 393073.179 | T | P | U | S_Nariño_2 | P_Nariño_2 | T |
| **Ext179** | UdenarStGua77 | CIP 393371.164 | T | P | U | S_Nariño_2 | P_Nariño_2 | T |
| **Ext180** | UdenarStGua62 | CIP 391046.14 | T | P | U | S_Nariño_2 | P_Nariño_2 | T |
| **Ext181** | UdenarStGua91 | CIP 396285.1 | T | P | U | S_Nariño_2 | P_Nariño_2 | T |
| **Ext182_1** | UdenarStGua66 | CIP 392285.72 | T | P | U | S_Nariño_2 | P_Nariño_2 | T |
| **Ext182_2 (Control)** | UdenarStGua66 | CIP 392285.72 | T | P | U | S_Nariño_2 | P_Nariño_2 | T |
| **Ext183** | UdenarStGua92 | CIP 397060.19 | T | P | U | S_Nariño_2 | P_Nariño_2 | T |
| **Ext184** | UdenarStGua72 | CIP 393079.24 | T | P | U | S_Nariño_2 | P_Nariño_2 | T |
| **Ext186** | UdenarStGua82 | CIP 394904.20 | T | P | U | S_Nariño_2 | P_Nariño_2 | T |
| **Ext187** | UdenarStGua59 | CIP 389746.2 | T | P | U | S_Nariño_2 | P_Nariño_2 | T |
| **Ext188** | UdenarStGua95 | CIP 398192.41 | T | P | U | S_Nariño_2 | P_Nariño_2 | T |
| **Ext189** | UdenarStGua64 | CIP 391580.30 | T | P | U | S_Nariño_2 | P_Nariño_2 | T |
| **Ext19** | UdenarStGua07 | Betina | T | C | Nariño | S_Nariño_2 | P_Nariño_2 | T |

(*Continued*)

**Table 1.** (Continued)

| Code | Genotype | Code or vulgar name | Ploidy by PD[1] | Country | Department | Pop_K2 [2] | Pop_K3 [3] | Ploidy by GS[4] |
|------|----------|---------------------|-----------------|---------|------------|-----------|-----------|-----------------|
| Ext190 | UdenarStGua73 | CIP 393079.4 | T | P | U | S_Nariño_2 | P_Nariño_2 | T |
| Ext191 | UdenarStGua93 | CIP 397196.3 | T | P | U | S_Nariño_2 | P_Nariño_2 | T |
| Ext192 | UdenarStGua75 | CIP 393280.82 | T | P | U | S_Nariño_2 | P_Nariño_2 | T |
| Ext193 | UdenarStGua74 | CIP 393220.54 | T | P | U | S_Nariño_2 | P_Nariño_2 | T |
| Ext194 | UdenarStGua97 | CIP 398193.553 | T | P | U | S_Nariño_2 | P_Nariño_2 | T |
| Ext195 | UdenarStGua88 | CIP 396034.103 | T | P | U | S_Nariño_2 | P_Nariño_2 | T |
| Ext196 | UdenarStGua71 | CIP 393077.159 | T | P | U | S_Nariño_2 | P_Nariño_2 | T |
| Ext198 | UdenarStGua85 | CIP 395438.1 | T | P | U | S_Nariño_2 | P_Nariño_2 | T |
| Ext199 | UdenarStGua81 | CIP 394895.7 | T | P | U | S_Nariño_2 | P_Nariño_2 | T |
| Ext20 | UdenarStGua27 | Nevada M6 | T | C | Nariño | S_Nariño_2 | P_Nariño_2 | T |
| Ext23 | UdenarStGua31 | Suprema Certificada M2 (Suprema) | T | C | Nariño | S_Nariño_2 | P_Nariño_2 | T |
| Ext236 | UdenarStGua35 | Guata Carriza M5 | T | C | Nariño | S_Nariño_2 | P_Nariño_2 | T |
| Ext237 | UdenarStGua37 | Morada Sigifredo M1 | T | C | Nariño | S_Nariño_2 | P_Nariño_2 | T |
| Ext238 | UdenarStGua25 | Chola Ecuatoriana | T | C | Nariño | S_Nariño_2 | P_Nariño_2 | T |
| Ext239 | UdenarStGua41 | San Pedro Invernadero | T | C | Nariño | S_Nariño_2 | P_Nariño_2 | T |
| Ext240 | UdenarStGua26 | Guata Negra Cordoba | T | C | Nariño | S_Nariño_2 | P_Nariño_2 | T |
| Ext247 | UdenarStCr49 | Monteña M1 | D | C | Nariño | S_Nariño_2 | P_Nariño_2 | T |
| Ext249 | UdenarStGua29 | Parda Pastusa Surco 22 M2 | T | C | Nariño | S_Nariño_2 | P_Nariño_2 | T |
| Ext25 | UdenarStGua28 | Pamba Lisa | T | C | Nariño | S_Nariño_2 | P_Nariño_2 | T |
| Ext27 | UdenarStGua34 | Capiro vieja M1 (Capiro Vieja La Cocha) | T | C | Nariño | S_Nariño_2 | P_Nariño_2 | T |
| Ext32 | UdenarStGua36 | Guata M5 Gualcala (Guata Gualcala) | T | C | Nariño | S_Nariño_2 | P_Nariño_2 | T |
| Ext39 | UdenarStGua22 | Guata 23 (Surco23) | T | C | Nariño | S_Nariño_2 | P_Nariño_2 | T |
| Ext43 | UdenarStGua10 | Guata parda | T | C | Nariño | S_Nariño_2 | P_Nariño_2 | T |
| Ext59 | Unknown | MAMA GUATA 22 M2 | U | U | U | S_Nariño_2 | P_Nariño_2 | T |
| Ext64 | UdenarStGua30 | Roja Nariño M1 | T | C | Nariño | S_Nariño_2 | P_Nariño_2 | T |
| Ext78 | UdenarStGua22 | Guata 23 (Surco23) | T | C | Nariño | S_Nariño_2 | P_Nariño_2 | T |
| Ext79 | UdenarStGua17 | Unica (Botana) | T | C | Nariño | S_Nariño_2 | P_Nariño_2 | T |
| Ext87 | UdenarStGua52 | 15062458—PEDIG-B 69S-76 XB-922-3 | T | U | U | S_Nariño_2 | P_Nariño_2 | T |
| Ext91 | UdenarStCr131 | Chaucha- 15061281 | D | C | Quindío | S_Nariño_2 | P_Nariño_2 | T |
| Ext10 | UdenarStGua40 | Roja Huila M6 | T | C | Nariño | S_Nariño_2 | P_Nariño_3 | T |
| Ext14 | UdenarStGua21 | Guata 21 (Surco21) | T | C | Nariño | S_Nariño_2 | P_Nariño_3 | T |
| Ext145 | UdenarStGua99 | CIP 399053.15 | T | P | U | S_Nariño_2 | P_Nariño_3 | T |
| Ext15 | UdenarStCr76 | nn vino tinto | D | C | Nariño | S_Nariño_2 | P_Nariño_3 | T |
| Ext155 | UdenarStCr177 | CIP 703545 | D | P | U | S_Nariño_2 | P_Nariño_3 | T |
| Ext157 | UdenarStGua60 | CIP 391002.6 | T | P | U | S_Nariño_2 | P_Nariño_3 | T |
| Ext178 | UdenarStGua56 | CIP 380496.6 | T | P | U | S_Nariño_2 | P_Nariño_3 | T |
| Ext18 | UdenarStGua23 | Guata 25 (Chola Surco 25) | T | C | Nariño | S_Nariño_2 | P_Nariño_3 | T |
| Ext185 | UdenarStCr181 | CIP 703508 | D | P | U | S_Nariño_2 | P_Nariño_3 | T |
| Ext217 | UdenarStCr80-1 | ju 11.2 | D | C | Nariño | S_Nariño_2 | P_Nariño_3 | T |
| Ext22 | UdenarStGua24 | Capiro Certificada M2 (Capiro) | T | C | Nariño | S_Nariño_2 | P_Nariño_3 | T |
| Ext234 | UdenarStCr20.1 | Curipamba 1.1 | D | U | U | S_Nariño_2 | P_Nariño_3 | T |
| Ext252 | UdenarStCr117 | Peruana- 15060543 | D | P | Cajamarca | S_Nariño_2 | P_Nariño_3 | T |
| Ext26 | UdenarStGua16 | Capiro Rosada | T | C | Nariño | S_Nariño_2 | P_Nariño_3 | T |
| Ext28 | Unknown | Mamá Capiro M6 | T | U | U | S_Nariño_2 | P_Nariño_3 | T |
| Ext41 | UdenarStGua13 | Bola de sal o pamba morada | T | C | Nariño | S_Nariño_2 | P_Nariño_3 | T |

(*Continued*)

**Table 1.** (Continued)

| Code | Genotype | Code or vulgar name | Ploidy by PD[1] | Country | Department | Pop_K2 [2] | Pop_K3 [3] | Ploidy by GS[4] |
|---|---|---|---|---|---|---|---|---|
| **Ext49** | UdenarStCr76 | nn vino tinto | D | C | Nariño | S_Nariño_2 | P_Nariño_3 | T |
| **Ext5** | UdenarStCr46 | Kurikinga M5 | D | C | Nariño | S_Nariño_2 | P_Nariño_3 | T |
| **Ext56** | UdenarStGua14 | Morasurco grande | T | C | Nariño | S_Nariño_2 | P_Nariño_3 | T |
| **Ext65** | UdenarStGua39 | Parda Suprema M6 | T | C | Nariño | S_Nariño_2 | P_Nariño_3 | T |
| **Ext7** | Unknown | San Juan Danita M2 | D | U | U | S_Nariño_2 | P_Nariño_3 | T |
| **Ext72** | UdenarStGua32 | Guata silvianaM1 (Guata Silvania La Cocha) | T | C | Nariño | S_Nariño_2 | P_Nariño_3 | T |
| **Ext77** | UdenarStGua33 | Capiro blanca M6 | T | C | Nariño | S_Nariño_2 | P_Nariño_3 | T |
| **Ext80** | UdenarStCr18 | C.I.O 35.16 | D | C | Nariño | S_Nariño_2 | P_Nariño_3 | T |
| **Ext88** | UdenarStCr166 | Chaucha Maleña- 15061755 | D | C | Nariño | S_Nariño_2 | P_Nariño_3 | T |
| **Ext9** | UdenarStGua09 | Leona | T | C | Nariño | S_Nariño_2 | P_Nariño_3 | T |
| **Ext92** | UdenarStGua45 | 15062413-Tocana blanca | T | C | Cundinamarca | S_Nariño_2 | P_Nariño_3 | T |

[1] Ploidy assigned according to Passport Data (PD)

[2] assignments determined through Bayesian analysis

[3] assignments determined through PCA

[4] ploidies assigned according to Genetic Structure (GS) analysis. D = Diploid; T = Tetraploid; U = Unknown; C = Colombia; P = Perú.

model implemented in the STRUCTURE program [27] without priori information for the population, evaluating one (K1) to ten (K10) possible subpopulations, with five independent repeats, assuming a mixture model with frequencies of correlated alleles investigated until 150,000 interactions. The optimal number of subpopulations was established with Evanno's method [28] in the Structure Harvester program [29] and in a model based on a Principal Component Analysis (PCA) carried out with the packages StAMPP [30] and Adegenet [31], where the number of subpopulations was determined with NBClust [32] and Factoextra [33] packages in the R program [26].

The number of subpopulations identified in each analysis was used to determine the genetic differentiation coefficients (FST) and percentages of differentiation between and within the subpopulations with Molecular Analysis of Variance (AMOVA) using the libraries StAMPP [30] and Poppr [34] in the R program [26]. The genetic diversity was estimated with observed heterozygosity (Ho), which was determined for each marker and each subpopulation based on the formula: Ho = Number total of heterozygous genotypes/Total number of genotypes (homozygous + heterozygous).

### Linkage disequilibrium

For the analysis of the linkage disequilibrium (LD) of the subpopulations detected in the potato genetic breeding collection at the Universidad de Nariño, the polymorphic SNP markers with a known physical position in the reference genome of *Solanum tuberosum* group Phureja DM1-3 used PGSC v4.03 Pseudomolecules [35]. Among the five possible genotypes for each marker (0, 1, 2, 3, 4), Pearson correlations ($r^2$) were calculated, and only the values with a level of significance lower than 0.001 were used to determine: 1) Linkage Disequilibrium (LD) averages at the subpopulation level and 2) how LD decays in genome plotting the $r^2$ values against physical distance in megabases (Mb), calculated between each combination of markers included in this analysis. These procedures were performed in the R program [26].

## Candidate genotypes for duplicates and/or possible use in controlled crosses

The identification of candidate genotypes for duplicates and/or possible use in controlled hybridization processes in the potato genetic breeding collection at the Universidad de Nariño was carried out through distribution of the Nei genetic distance [36], calculated in the StAMPP library [30] in the R program [26], for all genotypes included in the diploid and tetraploid subpopulations determined with the genetic structure analysis. Genotypes with genetic distances less than 0.010 were considered candidates for duplicates, while combinations of genotypes with genetic distances greater than 0.50 (in diploids) and 0.95 (in tetraploids) were selected as candidates for possible use in controlled crosses. For the identification of duplicates, the Ext_182 genotype (Table 1) was included as a control, which was genotyped in duplicate from two independent biological samples.

## Results

### Structure and genetic diversity

In the potato genetic breeding collection at the Universidad de Nariño, 4750 polymorphic SNP markers (57.2%) were identified, with an average of 340 markers per chromosome, distributed as follows: Chr 0 (84); Chr 1 (501); Chr 2 (433); Chr 3 (388); Chr 4 (502); Chr 5 (366); Chr 6 (403); Chr 7 (440); Chr 8 (353); Chr 9 (372); Chr 10 (258); Chr 11 (318); Chr 12 (268) and unanchored (64), of which 4602 were mapped on the 12 chromosomes of the potato genome. With all the polymorphic markers in this collection, the Bayesian and PCA analyses detected two (K2) and three (K3) possible subpopulations, respectively (Fig 1A and 1B).

The Bayesian analysis implemented in the STRUCTURE program for the potato genetic breeding collection at the Universidad de Nariño revealed that two (K2) clearly differentiated subpopulations were detected in an ACP barplot, which showed 34.76% of the genetic variability (Fig 2A), with an ancestry diagram (Fig 2B) that showed the genetic identity of each genotype in each identified group. The two subpopulations S_Nariño_1 and S_Nariño _2 made up

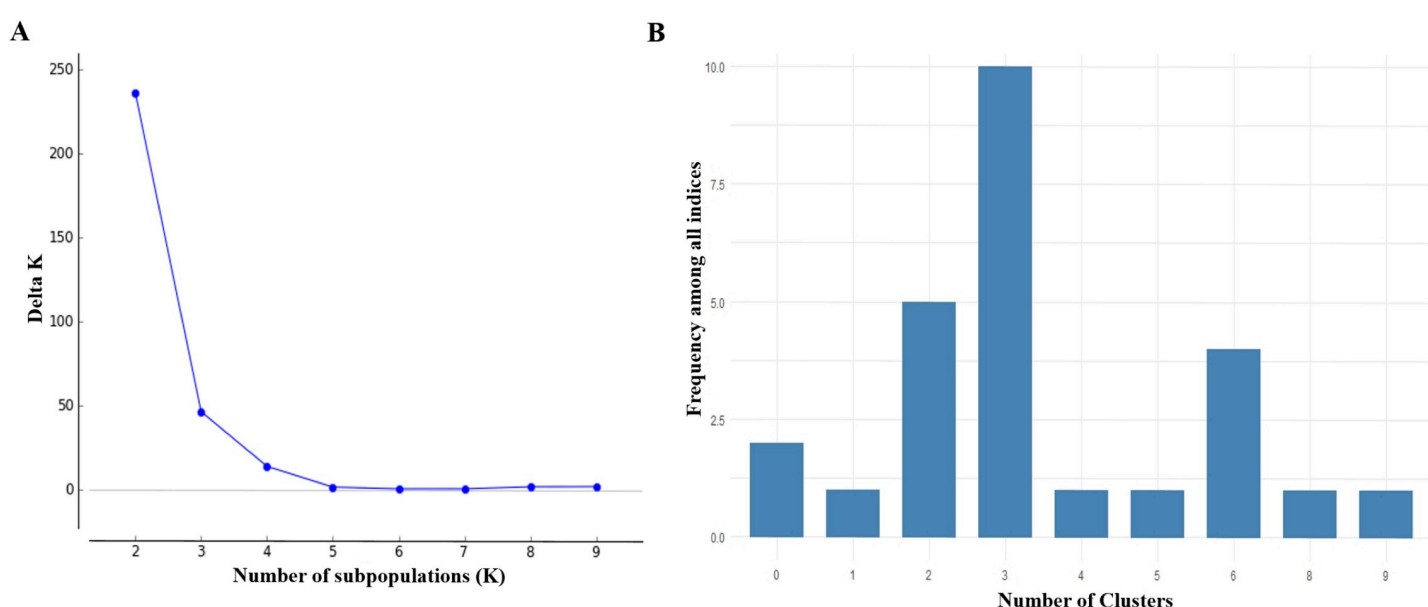

**Fig 1. Identification of the number of subpopulations in the potato breeding collection of the Universidad de Nariño.** A) Bayesian analysis; B) PCA.

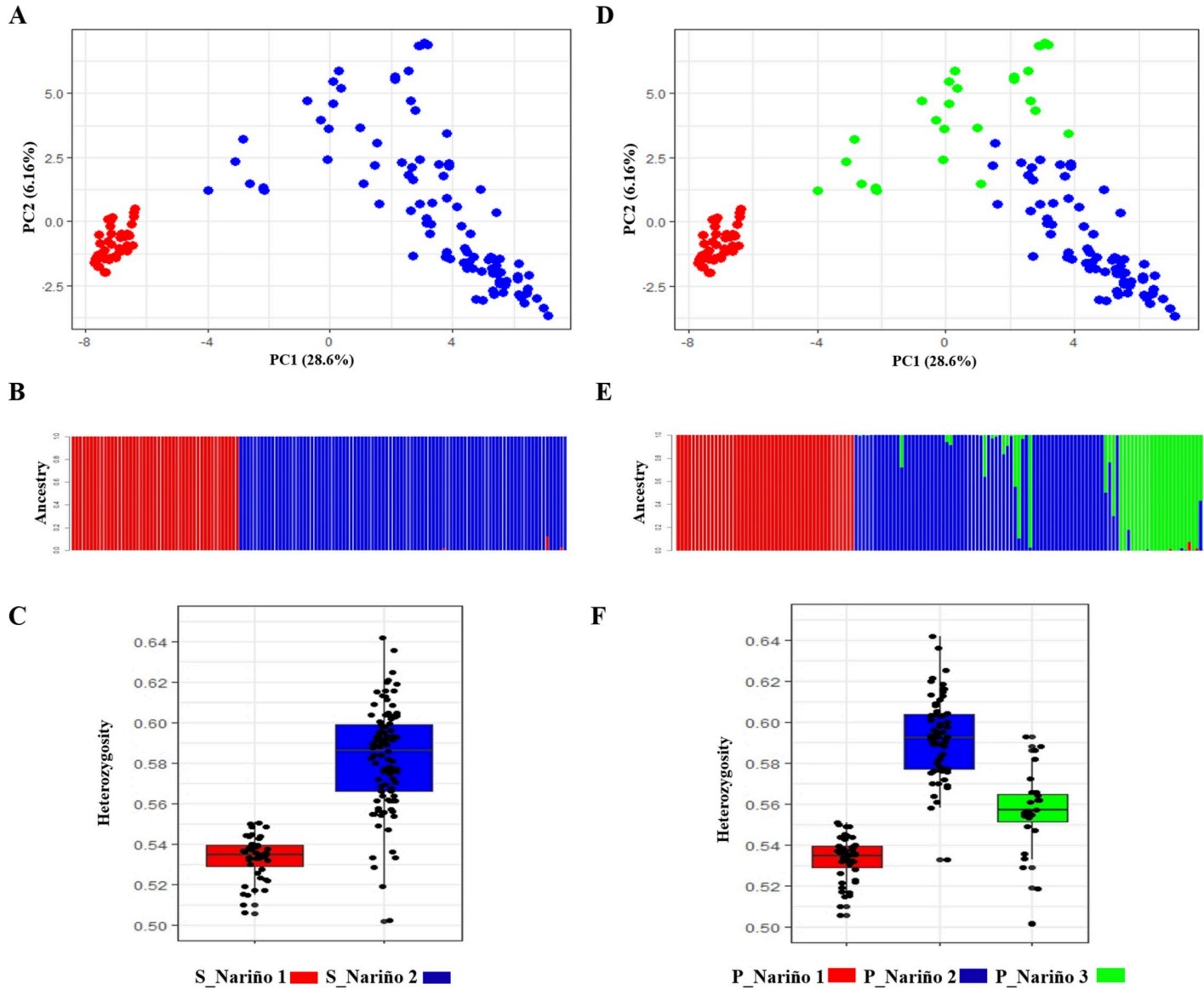

**Fig 2. Genetic analysis of the potato breeding collection of the Universidad de Nariño for the two (K2) and three (K3) subpopulations determined through Bayesian and PCA methods.** A) PCA K2; B) STRUCTURE barplot K2; C) Heterozygosity K2; D) PCA K3; E) STRUCTURE barplot K3; F) Heterozygosity K3.

of 47 and 97 genotypes in the high genetic differentiation, with an $F_{ST}$ between 0.533 and 63.31% between the populations and with high levels of heterozygosity (Ho> 0.53 and 36.69% of differentiation within the subpopulations), which was higher in the S_Nariño_2 subpopulation (Ho = 0.58) than in S_Nariño_1 (Ho = 0.53) (Table 2 and Fig 2C).

The samples grouped in subpopulation S_Nariño_1 were mainly (80%) from the Department of Nariño in Colombia, and the remaining samples (20%) were from Peru or had unknown origin. According to the passport data, the samples from this group mainly (91.5%) corresponded to diploid genotypes (43). However, four (8.5%) Colombian genotypes (Ext21, Ext48, Ext67 and Ext8) had passport data for tetraploids and/or were unknown (Table 1). On the other hand, subpopulation S_Nariño_2 had samples from Peru (54%), Colombia (40%) or unknown origin (6%), where 82.5% of the genotypes (80) had tetraploid passport data, while

**Table 2. Statistics of diversity, genetic structure, and Linkage Disequilibrium (LD) in the two (K2) and three (K3) subpopulations determined in the potato breeding collection of the Universidad de Nariño.**

| Analysis | Subp | PGS[1] | NS | Ho (M/R) | LD[2] (M/R) | AMOVA | | |
|---|---|---|---|---|---|---|---|---|
| | | | | | | FV | (%) | $F_{TS}$ Total |
| **STRUCTURE K2** | **S_Narino_1** | D | 47 | 0.53 (0.51–0.55) | 0.633 (0.47–1) | AP | 63.31 | 0.533* |
| | **S_Narino_2** | T | 97 | 0.58 (0.50–0.64) | 0.437 (0.33–0.99) | WP | 36.69 | |
| | **TOTAL** | | 144 | - | - | - | 100 | - |
| | | | | | | | | |
| **PCA K3** | **P_Narino_1** | D | 47 | 0.53 (0.51–0.55) | 0.633 (0.47–1) | AP | 44.78 | 0.536* |
| | **P_Narino_2** | T | 70 | 0.59 (0.53–0.64) | 0.510 (0.39–0.99) | WP | 55.22 | |
| | **P_Narino_3** | T | 27 | 0.56 (0.50–0.59) | 0.730 (0.60–0.99) | - | - | |
| | **TOTAL** | | 144 | - | - | - | 100 | - |

PGS[1] = Ploidy assigned according to Genetic Structure (GS) analysis; LD[2] (M/R) = Linkage disequilibrium (Mean/Range)

* = Significant at $p < 0.001$; NS = number of samples; Subp = subpopulations; Ho (M/R) = Heterozygosity (Mean/Range); D = diploids; T = tetraploids; AP = among populations; WP = within populations.

15 (10.4%) genotypes were Colombian, Peruvian or unknown (Ext105, Ext247, Ext59, Ext91, Ext15, Ext155, Ext185, Ext217, Ext234, Ext252, Ext49, Ext5, Ext7, Ext80 and Ext88), with diploid and/or unknown data (Table 1). According to the genetic analyses, this collection had 19 (13.2%) errors identified in the classification of genotypes according to level of ploidy. Thus, S_Nariño_1 and S_Nariño_2 were made up of possible diploid genotypes (2n = 2x = 24) and tetraploid genotypes (2n = 4x = 48), respectively (Table 1).

The analysis of the genetic breeding collection at the Universidad de Nariño based on ACP separated the two subpopulations of diploids (S_Nariño_1) and tetraploids (S_Nariño_2) detected with the Bayesian analysis in three (K3) possible subpopulations with the 47 (P_Nariño_1), 77 (P_Nariño_2) and 27 (P_Nariño_3) genotypes. This analysis also differentiated the diploid samples (S_Nariño_1 = P_Nariño_1) from the tetraploids (S_Nariño_2) and separated the latter into two subgroups, generating the subpopulations P_Nariño_2 and P_Nariño_3. The three subpopulations had a clear genetic differentiation with a $F_{ST}$ of 0.536 and 44.78% of differentiation between the populations (Fig 2D and 2E and Table 2) with high levels of heterozygosity, with Ho > 0.53 and 55.22% differentiation within the subpopulations, values that were higher in subpopulation P_Nariño_2 (Ho = 0.59), followed by P_Nariño_3 (Ho = 0.56) and P_Nariño_1 (Ho = 0.53) (Table 2 and Fig 2F). According to the passport data, subpopulation P_Nariño_2 was mainly made up of samples from Peru (66%), and P_Nariño_3 mainly had samples from Colombia (67%).

## Linkage disequilibrium of the potato breeding collection

The 4602 polymorphic markers mapped on the 12 chromosomes of the potato genome were used to evaluate the linkage disequilibrium (LD) in the two (K2) and three (K2) subpopulations detected in the potato breeding collection at the Universidad de Nariño, characterized by high levels of LD ($r^2 > 0.437$). For the K2 analysis, the LD levels were higher in subpopulation S_Nariño_1 ($r^2$ diploid = 0.633) than in S_Nariño_2 ($r^2$ tetraploid = 0.437), while in the K3 analysis, the two subpopulations of tetraploid genotypes detected in S_Nariño_2 had differences in the LD levels, which were higher in subpopulation P_Nariño_3 ($r^2$ Colombia = 0.730) than in P_Nariño_2 ($r^2$ Peru = 0.510) (Table 2). Indeed, LD, at a distance of approximately 3Mb, decayed slowly through the genome in all subpopulations detected for K2 and K3. The LD decayed at that distance with $r^2$ values of 0.63 in the diploid genotypes (S_Nariño_1 = P_Nariño_1) and 0.35 in the tetraploids (S_Nariño_1). In the tetraploid subpopulations

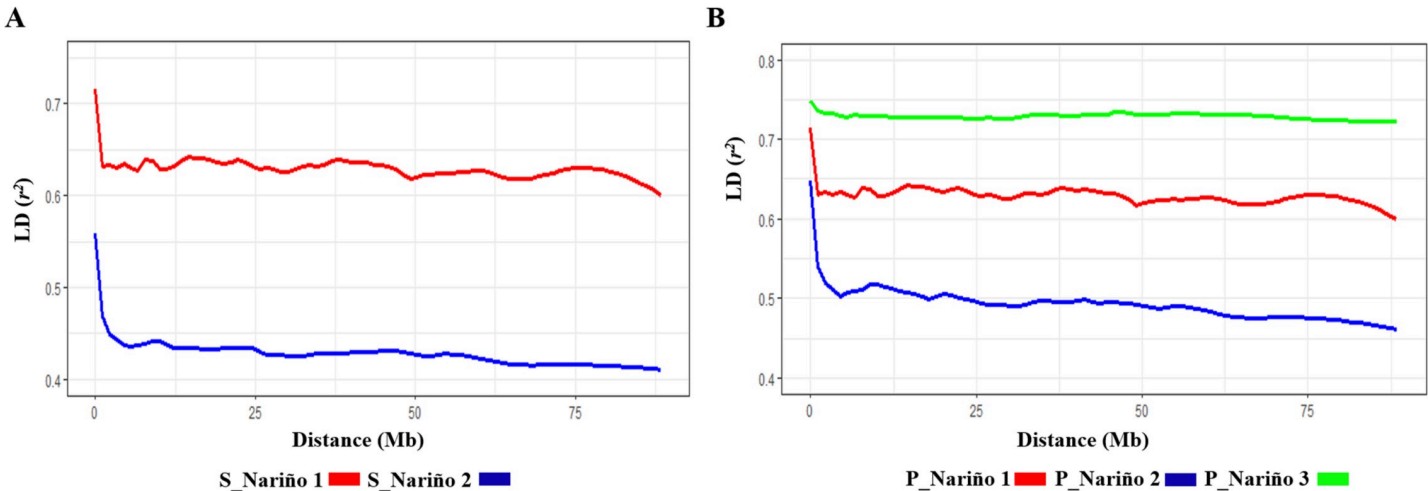

**Fig 3. Linkage Disequilibrium (LD) analysis for the two (K2) and three (K3) subpopulations determined in the potato breeding collection of the Universidad de Nariño.** A) LD K2 STRUCTURE; B) LD K3 PCA.

P_Nariño_2 and P_Nariño_3, the LD decayed at approximately 3Mb with an $r^2$ of 0.52 and 0.73, respectively (Fig 3A and 3B).

## Candidates to duplicates and crossing

The genetic distances between the samples that make up the potato genetic breeding collection at the Universidad de Nariño had a range from 0 to 0.110. These distances were greater in tetraploid genotypes S_Nariño_2 (mean of 0.065 and between 0 and 0.110) than in the diploid S_Nariño_1 (mean of 0.031 and between 0 and 0.056) (Fig 4A and 4B). The analysis of the diploid and tetraploid genotypes identified 25 possible candidates for duplicates with genetic

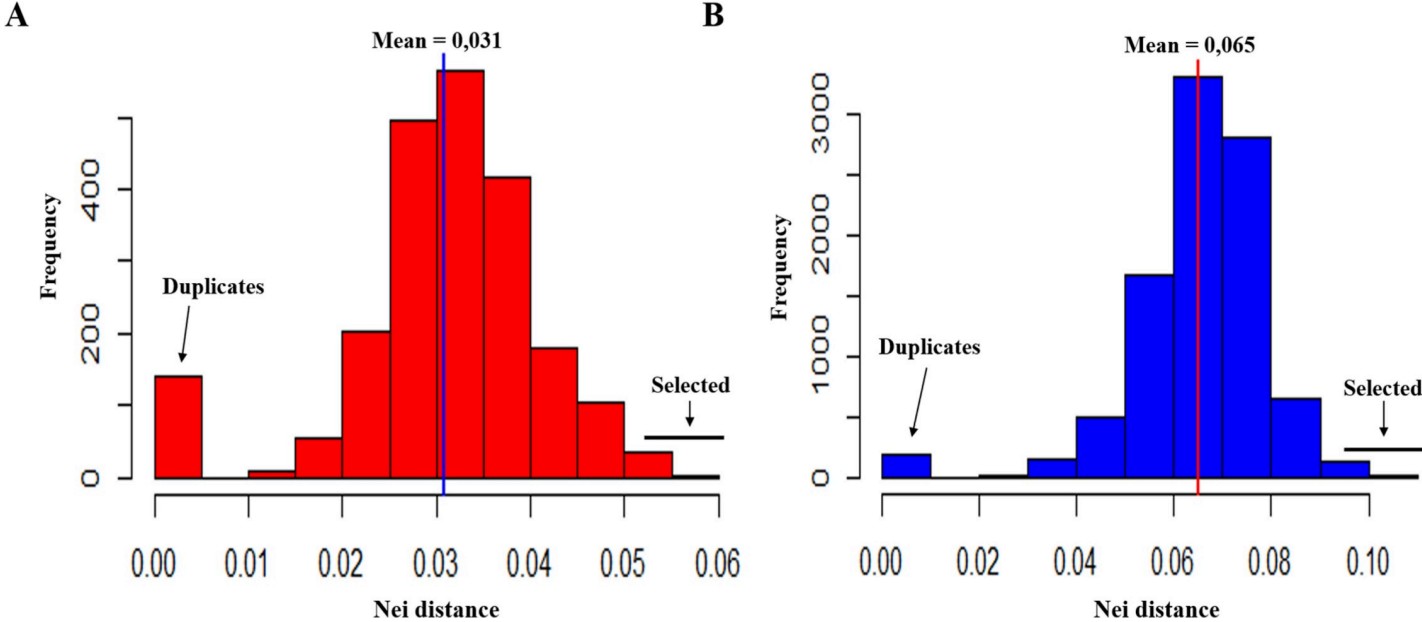

**Fig 4. Distributions of Nei genetic distances in the subpopulations of the potato breeding collection of the Universidad de Nariño.** A) Diploids (S_Nariño_1) genotypes; B) Tetraploids (S_Nariño_2) genotypes.

distances less than 0.01, including control duplicate 25, which corresponded to the identical samples Ext_182_1 and Ext_182_2. Additionally, 14 possible genotype combinations were identified in the diploid and tetraploid subpopulations because they had genetic distances greater than 0.50 and 0.95, respectively. The genotype combinations identified here can be used to implement controlled crosses in this collection (Table 3).

**Table 3. Genotypes of the potato breeding collection of Universidad de Nariño selected as candidates for duplicates and possible use in controlled crossing.**

| Selection | Populations | Number of duplicates or Crosses | Candidates for duplicates and crosses |
|---|---|---|---|
| Duplicates | S_Nariño_1 (Diploids) | Duplicate_1 | Ext1, Ext17, Ext3, Ext44, Ext67 |
| | | Duplicate_2 | Ext12, Ext21, Ext42, Ext48 |
| | | Duplicate_3 | Ext158, Ext229 |
| | | Duplicate_4 | Ext2, Ext45 |
| | | Duplicate_5 | Ext243, Ext250, Ext46 |
| | | Duplicate_6 | Ext245, Ext73 |
| | | Duplicate_7 | Ext246, Ext52 |
| | | Duplicate_8 | Ext33, Ext40 |
| | | Duplicate_9 | Ext4, Ext8, Ext68 |
| | | Duplicate_10 | Ext57, Ext62 |
| | S_Nariño_2 (Tetraploids) | Duplicate_11 | Ext11, Ext59 |
| | | Duplicate_12 | Ext150, Ext173 |
| | | Duplicate_13 | Ext164, Ext240 |
| | | Duplicate_14 | Ext166, Ext199 |
| | | Duplicate_15 | Ext167, Ext170 |
| | | Duplicate_16 | Ext174, Ext181 |
| | | Duplicate_17 | Ext179, Ext190 |
| | | Duplicate_18 | Ext20, Ext79 |
| | | Duplicate_19 | Ext239, Ext78 |
| | | Duplicate_20 | Ext247, Ext91 |
| | | Duplicate_21 | Ext14, Ext22, Ext26, Ext28 |
| | | Duplicate_22 | Ext15, Ext49, Ext72 |
| | | Duplicate_23 | Ext157, Ext65 |
| | | Duplicate_24 | Ext77, Ext92 |
| | | Duplicate_25 (Control) | Ext182_1, Ext182_2 |
| Crosses | S_Nariño_1 (Diploids) | Crosse_1 | Ext2 **X** Ext162, Ext214, Ext216, Ext229, Ext243, Ext245, Ext248, Ext4, Ext55, Ext8 |
| | | Crosse_2 | Ext214 **X** Ext2 |
| | | Crosse_3 | Ext216 **X** Ext2 |
| | | Crosse_4 | Ext229 **X** Ext2, Ext45 |
| | | Crosse_5 | Ext243 **X** Ext2 |
| | | Crosse_6 | Ext245 **X** Ext2, Ext45, Ext46 |
| | | Crosse_7 | Ext248 **X** Ext2 |
| | | Crosse_8 | Ext4 **X** Ext2, Ext45, Ext46 |
| | | Crosse_9 | Ext45 **X** Ext162, Ext229, Ext245, Ext4, Ext8 |
| | | Crosse_10 | Ext46 **X** Ext162, Ext245, Ext4, Ext8 |
| | | Crosse_11 | Ext55 **X** Ext2 |
| | | Crosse_12 | Ext8 **X** Ext2, Ext45, Ext46 |
| | S_Nariño_2 (Tetraploids) | Crosse_13 | Ext155 **X** Ext152, Ext160, Ext161, Ext166, Ext174, Ext177, Ext181, Ext182, Ext183, Ext187, Ext191, Ext192, Ext199, Ext87 |
| | | Crosse_14 | Ext185 **X** Ext152, Ext160, Ext161, Ext166, Ext174, Ext177, Ext181, Ext182, Ext183, Ext187, Ext191, Ext192, Ext199, Ext87 |

## Discussion

Genetic variability is crucial for the development of new cultivars with characteristics that the market requires, such as genotypes with resistance to diseases and/or pests, higher yields, quality and high nutritional values. Therefore, germplasms must be evaluated to identify new genetic sources with potential use in genetic breeding processes. In Colombia, the Department of Nariño has established itself as one of the main potato producers. However, the selection and/or generation of new cultivars adapted to the agroecological conditions of this region could increase the competitiveness of this department in domestic potato production. The potato genetic breeding collection at the Universidad de Nariño was evaluated at the genetic level based on molecular markers to establish parameters related to diversity, genetic structure, and linkage disequilibrium. This information is needed for the identification of candidate genotypes for duplicates and/or with potential use in genetic breeding processes.

The potato genetic breeding collection at the Universidad de Nariño consisted mainly of diploid and tetraploid genotypes originating from the Department of Nariño, known as a center of potato genetic diversity in Colombia [19] and also have genotypes from two of the more diverse genebanks for this specie, i.e. the CIP of Peru [37] and the CCC of Colombia [38]. This breeding collection is undergoing a morpho-agronomic evaluation under field conditions in different locations in the Department of Nariño to identify promising genotypes for the selection and/or development of new varieties that present outstanding attributes, such as high yield, good agro-industrial aptitude, and tolerance to diseases and abiotic stresses.

The collection at the Universidad de Nariño was analyzed with the 8303 SNPs included in the SNParray of SolCAP version 1 [16] to select genotypes, with a polymorphism level of 57.2%. The same panel of SNPs has been used to evaluate different potato populations with multiple origins. Berdugo-Cely et al. [19] identified 72% polymorphism among 809 diploid and tetraploid genotypes from the CCC in Colombia. Endelman et al. [39] identified 61% among 719 tetraploid genotypes from the United States. Esnault et al. [40] identified 61% among 48 tetraploid genotypes from the National Institute for Agronomic Research—INRA in France. Kolech et al. [41] identified 44.5% among 109 tetraploid genotypes from the United States, Europe, Peru and Ethiopia. Hardigan et al. [20] identified 61% among 287 diploid, tetraploid and hexaploid genotypes belonging to various species of *Solanum* sect. Petota and elite genotypes from the United States. Hirsch et al. [17] identified 77% among 250 monoploid, diploid, and tetraploid genotypes from the United States, and Stich et al. [18] identified 74% among 44 diploid and tetraploid genotypes of varieties grown in Europe. The differences in the percentage of polymorphism between the different studies is related to the number of samples used for comparison in studies that analyzed between 44 [18] and 809 [19] samples, with different levels of ploidy that included genotypes from monoploids [17] to hexaploids [20]. The high number of polymorphic markers identified in this study suggested that the SolCAP 8K matrix is suitable for the genetic analysis of the potato breeding collection at the Universidad de Nariño in Colombia.

The analysis of the population structure of the potato genetic breeding collection at the Universidad de Nariño based on the Bayesian analyses of the STRUCTURE and PCA program identified two and three possible subpopulations associated with the ploidy level, where diploid genotypes separated from tetraploids, and, according to the geographical origin, the tetraploid genotypes of Colombia separated from those of Peru. Multiple studies have described the use of molecular markers to classify and separate potato genetic materials conserved in germplasm banks according to their ploidy level [17–19, 42, 43] and the degree of genetic breeding to discriminate materials according to the varieties, cultivars, elite materials, wild species and/or related species [17, 42, 44–46].

The difference in the number of subpopulations identified between the two methods implemented in this study was related to their statistical bases. The STRUCTURE program identifies groupings with explicit genetic models for multiple population genetic parameters, which are often difficult to verify and require a lot of computing time and computational capacity [47, 48]. On the other hand, cluster analyses based on PCA identify genetic structures in large data sets with low computational capacity and shorter analysis times and do not use genetic models as a basis for identification. However, this alternative does not analyze a range of the number of populations and requires a priori definition of the number of populations to be detected. Additionally, it does not include all the information that STRUCTURE does since it summarizes the genetic variability of analyzed materials in a low number of components [47, 48]. However, it is one of the more commonly used methods for the evaluation of genetic structures in plant populations.

Multiple errors in the classification according to the ploidy level of the genotypes present in the potato genetic breeding collection at the Universidad de Nariño were identified in the tetraploid samples. The errors reported here must be confirmed with strategies such as flow cytometry, which will allow accurate corroboration of the ploidy in these genotypes. Errors in the genetic integrity of germplasm bank materials and genetic breeding collections conserved in field and *in vitro* conditions resulting from seed mixing, incorrect labeling, and errors in the data for origin and pedigree of the samples can be detrimental to genetic breeding programs [43]. However, these errors can be identified and adjusted with the support of a genetic analysis based on molecular markers, as reported in this study. Errors and adjustments in classifications according to ploidy levels [19, 43] and pedigree [39] of potato genotypes conserved in germplasm banks have been reported.

The diploid and tetraploid populations identified in the potato genetic breeding collection at the Universidad de Nariño had high levels of genetic diversity and linkage disequilibrium (LD) among the markers. The level of genetic diversity was lower in the diploid genotypes than in the tetraploids. At the LD level, differences were identified between the diploid genotypes and the tetraploids from Peru and those from Colombia. The tetraploid genotypes from Peru had greater genetic diversity than those from Colombia, while the genotypes from Colombia had higher levels of LD than those from Peru. Likewise, the LD decayed slowly in the potato genome of the diploid and tetraploid genotypes. In the tetraploids, the LD decayed slower in the genotypes from Colombia than in those from Peru. High values of heterozygosity [17, 19, 40, 42], and LD [17, 19, 40, 49, 50] have been reported in potato germplasm with the use of SNP markers, where diploid genotypes are characterized by a lower genetic diversity [19, 42–44] and higher levels of LD than in tetraploid genotypes [19]. Others diploid Colombian potato collections have been analyzed using SSR [51] and SNP markers [19] identifying high heterozygosity levels. High levels of heterozygosity in potatoes have been mainly associated with its heterozygous nature, allogamy, and broad variability in ploidy levels [52]. The differences between diploid and tetraploid potatoes in the heterozygosity levels has been associated with the ploidy bias, being higher these parameters in polyploid genotypes [53]. However, in this analysis to eliminate this bias all genotypes were analyzed as tetraploids, identifying a minor proportion of heterozygosity levels in diploid genotypes. On the other hand, the differences in the levels of genetic diversity and LD between the tetraploid genotypes from Colombia and Peru could be due to the fact that Peru is the center of origin for this species [52] and the fact that many of the samples analyzed here have not undergone strong selection.

In the potato genetic breeding collection at the Universidad de Nariño, candidates for duplicates and combinations of genotypes with broad genetic distances were identified that can be used to implement controlled crosses to generate populations with a high degree of heterosis. Candidates for duplicates included the Ext_182 control, indicating the reliability of the

genetic identity of the proposed duplicates and suggests that the SolCAP 8K chip [16] is a potential tool for the identification of duplicates in potato genotypes preserved and used in germplasm banks and/or breeding collections. Likewise, Kolech et al. [41] evaluated 44 potato genotypes grown in Ethiopia with the 8K chip and identified only 15 unique genetic materials, most of which were duplicate genotypes. The candidates for duplicates reported here must be validated with highly heritable morphological characteristics, such as shape and color of tubers and flowers, variables with high discriminatory power in potato germplasms at the morphological level [19, 54–56]. Errors in classification according to the level of ploidy and taxonomy and the presence of duplicate genotypes in germplasm banks and genetic breeding collections can be detrimental at an economic level in conservation strategies and for the selection of promising genotypes because they can identify materials with full genetic identity. Therefore, these materials must be identified and excluded for the estimation and identification of duplicates with molecular markers rather than conserving and using a duplicate accession as a different accession in a germplasm bank [57].

Genetic analyses with molecular markers can facilitate and support genetic breeding programs since they correct errors that occur in different stages, such as seed mixing and incorrect labeling, and establish genetic breeding strategies through the identification of materials and candidates for use in controlled breeding processes. It has been reported that one of the most important decisions in genetic breeding programs is the selection of the most suitable genotype for carrying out crosses that generate progeny with an increase in genetic gain [58]. The diploid and tetraploid genotypes selected according to levels of diversity and genetic distance for controlled crossing strategies identified in this study can be a baseline for possible genetic breeding strategies to be implemented with the germplasm from this collection. However, these genotypes must be verified with a morpho-agronomic characterization to establish their potential use.

## Conclusions

In the potato genetic breeding collection at the Universidad de Nariño in Colombia, high levels of heterozygosity were identified with a clear genetic structure that was mainly associated with the level of ploidy, which separated the diploid and tetraploid genotypes, discriminated the tetraploid genotypes, and differentiated the genotypes from Colombia and Peru. The genetic diversity was greater in the tetraploid genotypes than in the diploid genotypes. The tetraploid genotypes from Peru were more diverse than those from Colombia. The LD level was higher in the diploid genotypes than in the tetraploid genotypes, where the tetraploid genotypes from Colombia had higher LD levels than those from Peru. Multiple errors in the classification and candidates for duplicates in the potato breeding collection according to the level of ploidy were identified and adjusted. In the diploid and tetraploid genotypes, different combinations of candidate genotypes were identified for duplicates and/or for potential use in controlled hybridization processes. The genotype candidates for duplicates with errors in classification and/or potential use in future crosses must be validated with morpho-agronomic characterizations and flow cytometry. All results reported in this study suggested that the potato genetic breeding collection at the Universidad de Nariño has a broad genetic base with potential use for the genetic breeding of this crop in the Department of Nariño in southern Colombia.

## Supporting information

**S1 Table. Genotypic data of 144 accessions of potato breeding collection of Universidad de Nariño obtained through 8K SNParray technology.**
(XLSX)

## Acknowledgments

The authors thank the germplasm banks at the Centro Internacional de la Papa, the Colección Central Colombiana de Papa (AGROSAVIA) of Sistema de Bancos de Germoplasma de la Nación para la Alimentación y la Agricultura (SBGNAA), and the Universidad de Nariño for providing the genetic resources analyzed in this study, the Gobernación de Nariño, Universidad de Nariño and AGROSAVIA for their contribution to the structuring and approval of the project that financed this research.

## Author Contributions

**Conceptualization:** Carolina Martínez-Moncayo, Tulio César Lagos-Burbano.

**Data curation:** Jhon A. Berdugo-Cely.

**Formal analysis:** Jhon A. Berdugo-Cely.

**Funding acquisition:** Tulio César Lagos-Burbano.

**Investigation:** Jhon A. Berdugo-Cely.

**Methodology:** Jhon A. Berdugo-Cely, Carolina Martínez-Moncayo.

**Project administration:** Carolina Martínez-Moncayo, Tulio César Lagos-Burbano.

**Validation:** Jhon A. Berdugo-Cely.

**Visualization:** Jhon A. Berdugo-Cely.

**Writing – original draft:** Jhon A. Berdugo-Cely.

**Writing – review & editing:** Jhon A. Berdugo-Cely, Tulio César Lagos-Burbano.

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
