## [Decision Letter · Decision Letter 0]

24 Feb 2021

PONE-D-21-04514

Genetic analysis of a potato (Solanum tuberosum L.) breeding collection for southern of Colombia using single nucleotide polymorphism (SNP) markers

PLOS ONE

Dear Dr. Berdugo-Cely,

Thank you for submitting your manuscript to PLOS ONE. After careful consideration, we feel that it has merit but does not fully meet PLOS ONE’s publication criteria as it currently stands. Therefore, we invite you to submit a revised version of the manuscript that addresses the points raised during the review process.

We look forward to receiving your revised manuscript.

Kind regards,

Tzen-Yuh Chiang

Academic Editor

PLOS ONE

Journal Requirements:

2. During your revisions, please note that a simple title correction is required: to follow correct English language usage, the title should read "Genetic analysis of a potato (Solanum tuberosum L.) breeding collection for southern Colombia using single nucleotide polymorphism (SNP) markers". Please ensure this is updated in the manuscript file and the online submission information.

Reviewers' comments:

Reviewer's Responses to Questions

**Comments to the Author**

1. Is the manuscript technically sound, and do the data support the conclusions?

Reviewer #1: Partly

2. Has the statistical analysis been performed appropriately and rigorously? 

Reviewer #1: I Don't Know

3. Have the authors made all data underlying the findings in their manuscript fully available?

Reviewer #1: Yes

4. Is the manuscript presented in an intelligible fashion and written in standard English?

Reviewer #1: Yes

5. Review Comments to the Author

Reviewer #1: Needs to acknowledge and incorporate related work before further review...

Juyó, D., F. Sarmiento, M. Álvarez, H. Brochero, C. Gebhardt, T. Mosquera. 2015. Genetic Diversity and Population Structure in Diploid Potatoes of Solanum tuberosum Group Phureja. Crop Science 55: 760-769.

Technical issue is to acknowledge and incorporate recent publications on ploidy bias, e.g.,...

Bamberg, J. and A. del Rio. 2020. Assessing under-Estimation of Genetic Diversity within Wild Potato (Solanum) Species Populations. American Journal of Potato Research 97:547-553

6. PLOS authors have the option to publish the peer review history of their article (what does this mean?). If published, this will include your full peer review and any attached files.

Reviewer #1: No

---

## [Author Response · Author response to Decision Letter 0]

2 Mar 2021

Dear Editor,

PLOS ONE

Here describe all answers for the editor and reviewers and the changes made in the manuscript,

Commentary editor

1. All references were cited in the list included the suggested by reviewers.

2. The title of the paper was adjusted being the editor commentary: “Genetic analysis of a potato (Solanum tuberosum L.) breeding collection for southern Colombia using single nucleotide polymorphism (SNP) markers” in the document and web submission on-line.

3. All manuscript was adjusted being PLOS ONE's style requirements (the format changes as line spacing, tables format and subtitle size was not included in track changes), the author affiliations were corrected because this was inverted to the two last authors.

Commentary reviewers

4. The references suggested was reviewed and included in the manuscript in the lines 340-341 “Others diploid Colombian potato collections have been analyzed using SSR [51] and SNP markers [19] identifying high heterozygosity levels.” and lines 343-346 “The differences between diploid and tetraploid potatoes in the heterozygosity levels has been associated with the ploidy bias, being higher these parameters in polyploid genotypes [53]. However, in this analysis to eliminate this bias all genotypes were analyzed as tetraploids, identifying a minor proportion of heterozygosity levels in diploid genotypes.”. I

5. This, these two new references were included: 51) Juyó D, Sarmiento F, Álvarez M, Brochero H, Gebhardt C, Mosquera T. Genetic Diversity and Population Structure in Diploid Potatoes of Solanum tuberosum Group Phureja. Crop Sci. 2015;55: 760–769. doi:https://doi.org/10.2135/cropsci2014.07.0524 and 53) Bamberg J, del Rio A. Assessing under-Estimation of Genetic Diversity within Wild Potato (Solanum) Species Populations. Am J Potato Res. 2020;97: 547–553. doi:10.1007/s12230-020-09802-3.

Additional changes

6. In the text was included: “The genetic materials in this collection belong from multiple germplasm bank origins: the International Potato Center (CIP) of Peru, the Colombian Central Collection (CCC), and the Universidad de Nariño of Colombia and” in the lines 105-107.

7. The sections of Acknowledgments, Funding were edited, while the Contributions section was deleted.

8. Acknowledgments: The authors thank the germplasm banks at the Centro Internacional de la Papa, the Colección Central Colombiana de Papa (AGROSAVIA) of Sistema de Bancos de Germoplasma de la Nación para la Alimentación y la Agricultura (SBGNAA), and the Universidad de Nariño for providing the genetic resources analyzed in this study, the Gobernación de Nariño, Universidad de Nariño and AGROSAVIA for their contribution to the structuring and approval of the project that financed this research.

9. Funding: The resources for the development of this research were provided by the Ministerio de Ciencia y Tecnología de Colombia (Minciencias) and the Gobernación de Nariño through the Fondo de Ciencia, Tecnología e Innovación del Sistema General de Regalias, with the approval of the project "Improvement technological and productive of the potato system in the Department of Nariño”, identified with code BPIN No. 2014000100022. The execution of this project was carried out between the Universidad de Nariño and AGROSAVIA through the macro agreement 480-15. The hours spent by researcher Jhon A. Berdugo-Cely MSc. for the development of this study were provided by AGROSAVIA through Variable Transfer (TV) 2019.

10. The S1Table was included as supporting information.

Best regards,

Jhon Berdugo

---

## [Editor Report · Decision Letter 1]

5 Mar 2021

Genetic analysis of a potato (Solanum tuberosum L.) breeding collection for southern Colombia using single nucleotide polymorphism (SNP) markers

PONE-D-21-04514R1

Dear Dr. Berdugo-Cely,

We’re pleased to inform you that your manuscript has been judged scientifically suitable for publication and will be formally accepted for publication once it meets all outstanding technical requirements.

Kind regards,

Tzen-Yuh Chiang

Academic Editor

PLOS ONE
---

## [Editor Report · Acceptance letter]

9 Mar 2021

PONE-D-21-04514R1 

Genetic analysis of a potato (*Solanum tuberosum* L.) breeding collection for southern Colombia using single nucleotide polymorphism (SNP) markers 

Dear Dr. Berdugo-Cely:

I'm pleased to inform you that your manuscript has been deemed suitable for publication in PLOS ONE. Congratulations! Your manuscript is now with our production department. 

Kind regards, 

on behalf of

Dr. Tzen-Yuh Chiang 

Academic Editor

PLOS ONE